# Sustained Activation of the Anterior Thalamic Neurons with Low Doses of Kainic Acid Boosts Hippocampal Neurogenesis

**DOI:** 10.3390/cells11213413

**Published:** 2022-10-28

**Authors:** Farah Chamaa, Batoul Darwish, Rami Arnaout, Ziad Nahas, Elie D. Al-Chaer, Nayef E. Saadé, Wassim Abou-Kheir

**Affiliations:** 1Department of Anatomy, Cell Biology and Physiological Sciences, Faculty of Medicine, American University of Beirut, Beirut P.O. Box 11-0236, Lebanon; 2Biological and Environmental Science and Engineering Division, King Abdullah University of Science and Technology, Thuwal 23955, Saudi Arabia; 3Department of Psychiatry, University of Minnesota, Minneapolis, MN 55455, USA

**Keywords:** kainic acid stimulation, neurogenesis, hippocampus, stem/progenitor cells, anterior nucleus, dexamethasone

## Abstract

Adult hippocampal neurogenesis is prone to modulation by several intrinsic and extrinsic factors. The anterior nucleus (AN) of the thalamus has extensive connections with the hippocampus, and stimulation of this region may play a role in altering neurogenesis. We have previously shown that electrical stimulation of the AN can substantially boost hippocampal neurogenesis in adult rats. Here, we performed selective unilateral chemical excitation of the cell bodies of the AN as it offers a more specific and sustained stimulation when compared to electrical stimulation. Our aim is to investigate the long-term effects of KA stimulation of the AN on baseline hippocampal proliferation of neural stem cells and neurogenesis. Continuous micro-perfusion of very low doses of kainic acid (KA) was administered into the right AN for seven days. Afterwards, adult male rats received 5′-bromo-2′-deoxyuridine (BrdU) injections (200 mg/kg, i.p) and were euthanized at either one week or four weeks post micro-perfusion. Open field and Y-maze tests were performed before euthanasia. The KA stimulation of the AN evoked sustained hippocampal neurogenesis that was associated with improved spatial memory in the Y-maze test. Administering dexamethasone prior to and simultaneously with the KA stimulation decreased both the hippocampal neurogenesis and the improved spatial recognition memory previously seen in the Y-maze test. These results suggest that hippocampal neurogenesis may be a downstream effect of stimulation in general, and of excitation of the cell bodies of the AN in particular, and that stimulation of that area improves spatial memory in rats.

## 1. Introduction

Adult neurogenesis is a dynamic process that leads to the formation of new neurons within restricted brain niches. Newly born cells in the dentate gyrus of the hippocampus are especially important for cognitive functions and memory encoding [1]. Despite its dynamic feature, neurogenesis can be modulated by multiple factors, including aging, neurodegenerative diseases, and environmental conditions [2,3]. There have been substantial efforts in the last few years to discover what could regulate, maintain, and even boost neurogenesis [4]. It is well established that exercise, positive social experience, and administration of certain vitamins and some anti-depressant drugs all boost neurogenesis [5,6,7,8]. We have previously demonstrated that electrical stimulation of the anterior nucleus (AN) of the thalamus, which has substantial connections with the hippocampus, significantly increases hippocampal neurogenesis [9,10]. Nonetheless, the mechanism is not yet clearly understood since electrical stimulation can be non-specific and can excite passing fibers, leading to diffused effects in nearby regions.

More specific targeting of the AN neurons, which offers a clearer interpretation at the cellular level, is the use of chemical stimulation [11]. Kainic acid (KA) is a potent neuro-excitatory amino acid that acts as a direct agonist to glutamic kainate receptors, a class of glutamate receptors. At larger doses, KA is considered a neurotoxin that induces seizure-like firing and can lead to neuronal death due to excitotoxicity [12,13,14,15,16]. However, diluted small doses of kainic acid are known to chemically stimulate neurons [11]. Unlike studies that deliver KA systematically, we opted for the local delivery of KA directly into the AN. Usually, KA injection in a specific area leads to excitation of neuronal cell bodies in that area without activating axons of passage [17].

Therefore, we opted to investigate whether exciting the cell bodies of the AN, using a low dose of kainic acid, would have an effect on neurogenesis. We were also interested in examining whether KA, in addition to enhancing baseline neurogenesis, could ameliorate the disruptions of neurogenesis that may occur due to pathological conditions, such as chronic stress or sustained increase in corticosteroid levels [18]. Thus, dexamethasone, a corticosteroid medication that is well documented to inhibit the proliferation of stem/progenitor cells in the DG of the hippocampus, was administered [19].

In this study, we performed continuous micro-perfusion of low doses of KA using osmotic mini-pumps to produce sustained activation of the AN neurons. We studied the long-term effects of the KA stimulation on baseline hippocampal neurogenesis on the one hand and on disrupted neurogenesis following dexamethasone injections on the other. We also investigated the association between the findings on neurogenesis and the effects on spatial memory and general motor behavior in adult rats.

## 2. Materials and Methods

### 2.1. Animals

In this study, male Sprague-Dawley rats (250–300 g) were housed in a controlled environment with a 12 h light/dark cycle where temperature was constantly set to 23 ± 2 °C and water and chow were provided ad libitum. All the experimental procedures were performed in accordance with the Institutional Animal Care and Use Committee (IACUC). All surgical procedures were conducted under deep anesthesia using intraperitoneal (*ip*) injection of ketamine (Ketalar^®^; 50 mg/kg) and xyla (Xylazine^®^; 12 mg/Kg). Postoperative surveillance of behavior and body weight were monitored daily during the experiment.

### 2.2. Experimental Groups

This study reports the observations made on five groups of adult rats that were subjected to the following treatments: Two groups were implanted with mini-osmotic pumps containing either KA (*n* = 6) or sterile saline (*n* = 6) and euthanized after 1 week (day 9) of micro-perfusion. Three groups were implanted with mini-pumps filled with either sterile saline (*n* = 7), KA (*n* = 7), or KA and daily treatment with dexamethasone (*n* = 6) over one week and were euthanized after 4 weeks of micro-perfusion (Figure 1).

### 2.3. Stereotactic Cannula Implantation

Under deep anesthesia and in an aseptic condition, the head of the rats was firmly fixed on a stereotaxic frame (DKI, Elk Grove Village, IL, USA). After exposing the skull, a hole was drilled above the targeted area, and a brain infusion cannula (Brain Infusion Kit 1, Alzet, Durect Corporation, Cupertino) was implanted in the right AN and then fixed on the skull with acrylic glue. The following stereotaxic coordinates were used: −1.4 mm caudal to the bregma; +0.8 mm lateral, with reference to the midline; and 6 mm vertical from the surface of the brain [20]. The implanted cannula was connected using a thin polyethylene catheter (PE-10) to a mini-osmotic pump, which was surgically inserted under the skin in the dorsolateral area of the abdomen of the rat [21].

### 2.4. Kainic Acid Administration

Kainic acid or saline was administered as continuous micro-perfusion using a mini-osmotic pump (Mini-Osmotic Pump, Model 2001, Alzet, Cupertino, CA, USA) at a low dose (500 pM given at 1µL/h injection) for 7 days. KA powder (Tocris Bioscience, Bristol, UK) was weighed and diluted in sterile saline at an initial concentration of 1 mg/ml and was further diluted to reach a final concentration of 500 pM in each 1 µl injection.

### 2.5. Dexamethasone Administration

The rats received daily intraperitoneal injections of dexamethasone (Sigmatec, North Royalton, OH, USA; 8 mg/2 mL ampoule), which was diluted in a saline solution (200 µg/kg/injection), 2 days prior to and during the period of KA micro-perfusion (a total of 9 days). The rats were euthanized after 4 weeks to assess the survival and differentiation of neural progenitor cells (Figure 1B).

### 2.6. BrdU Administration

BrdU powder (Sigma-Aldrich, B5002) was weighed and dissolved in 0.9% of warm sterile saline to reach a final concentration of 200 mg/kg, and injected at 900 µL into each rat. The rats received 3 BrdU injections (66 mg/kg/ 300 µl injection, ip), each separated by 2 h, on day 8 after one week of micro-perfusion with either KA or sterile saline (Figure 1).

### 2.7. Novel Arm Exploration

The Y-maze test was used to assess spatial memory in the rats. The apparatus consists of three identical arms (10 cm wide and 40 cm long) that are equally spaced (120° apart). No intra-maze cues were added, but different objects were placed at a range of distances outside the maze that were visible to the rats and served as extra-maze cues. Training and testing were performed, as previously described by our lab [9]. Briefly, the test consisted of two phases that were 1 h apart. In the first training, or the acquisition phase, one of the arms which was denoted as the novel arm was blocked and the rats were placed in the “Start” arm. A period of 15 min was timed for the rats to explore and familiarize themselves with both the “start” (S) and the “familiar” arms (F). In the second phase (5 min), or the test phase (retention trial), the closed/novel arm (N) was open and the rats were initially placed in the “S” arm. The test phase was video recorded. The Y-maze test was performed at two different time points: day 8 post micro-perfusion and day 31 post micro-perfusion. The sham and KA groups that were euthanized on day 31 also performed the Y-Maze test on day 8 and, thus, their data were combined.

### 2.8. Open Field

An open field test was used to assess locomotor activity and anxiety-associated behavior in the rats to determine whether the kainic acid micro-perfusion had an effect on the rats’ motor activity [22]. The rats were allowed 20 to 30 min to habituate to the experimental room, and then they were successively placed in the open field apparatus (no prior exposure to the apparatus) for a 5-min testing that was recorded by a camera. The videos were analyzed using the ANY-maze^TM^ behavioral tracking software. Multiple variables were recorded. The central zone was designated as 25% of the area of the open field apparatus [22,23].

### 2.9. Tissue Sampling and Processing

At the end of the observation period, individual rats were deeply anesthetized and perfused transcardially with a solution containing 0.9% of saline followed by 4% of formalin. The brains were carefully removed and fixed in 4% paraformaldehyde for one day, and then transferred to a 30% sucrose solution in 0.1 M PBS and stored at 4 °C until full impregnation occurred. Brain sections were cut using a sliding microtome and sampled in a systemic manner as 6 sets using the fractionator method, as previously described [9,24]. Briefly, 40 μm coronal sections were cut serially from the rostral to the caudal extent of the DG at the following rostro-caudal coordinates to cover the whole hippocampal formation: −2.12 to −6.3 mm relative to the bregma. To highlight the topographic correspondence of the BrdU distribution, the DG region was divided into three areas as follows: the rostral region ranging from −2.12 to −3.7 mm relative to the bregma; the intermediate region ranging from −3.7 to −4.9; and the caudal region ranging from −4.9 to −6.3 [20,25]. The rostral region is the dorsal hippocampus, whereas the intermediate and caudal regions are part of the ventral hippocampus. All sections were collected and stored in a sodium azide solution (15 mM in 0.1 M PBS).

### 2.10. Immunofluorescence and Confocal Microscopy

One well out of the 6 representative wells from each DG region (rostral, intermediate, and caudal) was randomly chosen from each rat and immunofluorescence was performed, as previously described [9]. Briefly, the sections were washed and incubated at 37 °C with 2N HCL to allow denaturing of DNA, then 0.1 M of sodium borate (pH 8.5) was added for 10 min. The tissues were incubated overnight with the following monoclonal primary antibodies: rat anti-BrdU (1:100; BioRad, Hercules, CA, USA) and monoclonal mouse anti-NeuN (1:500; Millipore) diluted in PBS with 3% of NGS, 3% of BSA, and 0.1% of Triton-X. Incubation with the secondary antibodies was performed on the following day for 2 h at room temperature on a shaker, using Alexa Fluor-568 anti-rat (1:200; Molecular Probes, Invitrogen, Eugene, OR, USA) and Alexa Fluor-488 anti-mouse (1:250; Molecular Probes, Invitrogen). Finally, Hoechst stain (1:10,000 in PBS; Molecular Probes, Invitrogen) was added to the sections before they were washed and mounted onto the slides using Fluoro-Gel without DAPI (Electron Microscopy Sciences, Hatfield, PA, USA).

### 2.11. Cell Counting

As described previously [9], one set of sections was randomly chosen per rat and cell counting of BrdU-positive cells was confined to the subgranular zone (SGZ) of the DG at week 1 and the granular cell layer (GCL) at week 4. BrdU+ cells were counted manually using a 40X-oil objective. The total number of positive cells counted per rat was multiplied by 6 (the number of sets per rat) to obtain the overall number of BrdU+ cells in each region (rostral, intermediate, and caudal) of the DG for each rat.

### 2.12. Microscopic Analysis

Microscopic analysis was performed using the Zeiss LSM 710 confocal microscope, and the images were acquired using a 40X-oil objective as Z stacks and a tile scan was performed to show how the BrdU+ cells were distributed within the 40 µm section of the whole dentate gyrus of each region. The images were analyzed using the Zeiss ZEN 2009 image-analysis software and they were processed with maximal intensity projection. For consistency, BrdU+ cells were counted by a single-blinded researcher, and the images were acquired under the same laser and the same microscopic parameters.

### 2.13. Statistical Analysis

Statistical analysis and plotting of figures were performed using the Prism 6 GraphPad package (GraphPad software, Inc., San Diego, CA, USA). The BrdU cell count data were presented as mean ± SEM. The following variables were taken into consideration: KA micro-perfusion (sham/KA), side of micro-perfusion (contralateral/ipsilateral), and region of DG (rostral/intermediate/caudal). The determination of the significance of differences was performed using a two-way ANOVA followed by Tukey’s or Sidak’s multiple comparison test, accordingly, and by unpaired *t*-test for the topographical distribution. A one-way repeated measures ANOVA followed by Tukey’s multiple comparison test was used to assess the statistical significance in the Y-maze test. A *p*-value of <0.05 was considered the limit of the significance of differences at 95% confidence interval.

## 3. Results

### 3.1. KA Micro-Perfusion Increased Dentate Gyrus Cell Proliferation after 8 Days

Micro-perfusion of the AN with low doses of KA induced a 1.5-fold increase in the number of BrdU-positive cells in both the contralateral and the ipsilateral (with reference to the site of injection) SGZ of the DG. The number of BrdU-positive cells counted in the KA micro-perfused group was significantly higher than that of the sham group (2081 ± 197 versus 1410 ± 84, *p* = 0.028, in the contralateral DG, and 2134 ± 111 versus 1449 ± 93, *p* = 0.025, in the ipsilateral DG, respectively) (F (1,16) = 20.35, *p* = 0.0004; Tukey’s multiple comparison test, *p* < 0.05) (Figure 2A,B).

BrdU-positive cells were examined along three areas of the DG: the rostral, intermediate, and caudal regions. KA micro-perfusion induced a significant increase that was observed only in the caudal region of the DG at both the contralateral (1203 ± 149 versus 672 ± 56 in the sham group; *p* = 0.0249, Figure 3A,B) and the ipsilateral sides (1482 ± 150 versus 712 ± 77 in the sham group; *p* = 0.0046, Figure 3A,B).

### 3.2. KA Micro-Perfusion Increased Neurogenesis after 4 Weeks

The total number of labeled cells elicited an almost 3-fold increase in BrdU-positive cells in both the contralateral DG (1730 ± 214 versus 424 ± 51 in the sham group; *p* < 0.001) and the ipsilateral DG (1869 ± 184 versus 378 ± 54 in the sham group) (F (1,18) = 76.61, *p* < 0.001; Sidak’s multiple comparison test, *p* < 0.001) (Figure 4A,C). However, the % of double-positive BrdU/NeuN cells out of the total BrdU-positive cells was similar between the sham and KA micro-perfused groups, and there were no differences within the same group between the contralateral (90.71% ± 1.48 and 92.9% ± 1.38 in the sham and KA micro-perfused groups, respectively) and the ipsilateral DG (88.42% ± 1.36 and 90.69% ± 2.99 in the sham and KA micro-perfused groups, respectively) (Figure 4B).

The highest increase was in the caudal region of the DG in both the ipsilateral and contralateral sides: from 198 ± 23 in the contralateral side of the sham group to 1198 ± 216 after KA micro-perfusion (*p* = 0.0025, Figure 5A,B), and from 171 ± 17 in the ipsilateral side of the sham group to 1246 ± 186 after KA micro-perfusion (*p* = 0.0006, Figure 5A,B).

The rats that were treated with dexamethasone two days prior to and during KA micro-perfusion had a total number of BrdU-positive cells at week 4 that was comparable to the sham group in both the contralateral (776 ± 77; Tukey’s multiple comparison test, *p* = 0.476) and ipsilateral DG (728 ± 79; Tukey’s multiple comparison test, *p* = 0.48) (Two-way ANOVA, F(2, 28) = 58.02, *p* < 0.0001; Appendix A).

### 3.3. Improved Spatial Memory in KA Micro-Perfused Group

In the Y-maze test that was performed at week 1, the sham rats spent comparable time exploring the three different arms of the Y-maze, with a higher tendency to explore the familiar arm (one-way repeated measures ANOVA, F(1.653, 19.84) = 4.568, *p* = 0.0289; Tukey’s multiple comparison test). However, the rats that were micro-perfused with KA showed significantly longer exploration of the novel arm (133.55± 11.78 s) compared to the familiar (85.53 ± 8.02 s, *p* = 0.0224) and start arms (76.36± 10.31 s, *p* = 0.0062) (one-way repeated measures ANOVA, F(2, 24) = 6.718, *p* = 0.0048; Tukey’s multiple comparison test) (Figure 6A).

Exploratory behavior of the rats at week 4 was almost comparable to their performance at week 1 (Figure 6B). The sham rats equally explored the three different arms (one-way repeated measures ANOVA, F (1.15, 6.898) = 2.368, *p* = 0.1688). The KA micro-perfused rats retained the same behavioral exploration performance, with more time spent in the novel arm (138.6 ± 11.2 s) compared to the familiar arm (79 ± 12.9 s, *p* = 0.0248) or the start arm (76.6 ± 9.8 s, *p* = 0.0373) (one-way repeated measures ANOVA, F(1.813, 10.88) = 6.833, *p* = 0.0134; Tukey’s multiple comparison’s test).

Dexamethasone injections given to the KA micro-perfused rats suppressed the increased tendency to explore the novel arm at week 1 and week 4. The rats showed a higher tendency to explore the familiar arm (138.3 ± 6.8 s) compared to the novel arm at week 1 (66.98 ± 2.79 s, *p* = 0.0096) (one-way repeated measures ANOVA, F(1.608, 4.825) = 28.7, *p* = 0.0024; Tukey’s multiple comparison test; Appendix A). After four weeks of daily treatment with dexamethasone, the KA micro-perfused rats maintained a reduced exploration of the novel arm (one-way repeated measures ANOVA, F (1.139, 4.557) = 2.478, *p* = 0.1837; Appendix A).

### 3.4. KA Micro-Perfusion Does Not Affect Motor and Exploratory Behaviors in the Open Field Test

In the open field test, the KA micro-perfused rats had comparable motor parameters to those exhibited by the sham rats. There were no significant differences in the total distance traveled, the average speed, and the total time spent immobile (Figure 7A). Moreover, the KA, sham, and KA+ Dexa groups all had a similar tendency to explore the central zone of the open field (Figure 7B).

## 4. Discussion

The present study used the technique of continuous release of low doses of kainic acid to activate neurons of the anterior thalamic nuclear group of the thalamus, in an attempt to simulate the activation of these neurons under physiological conditions. Unilateral micro-perfusion of the AN with KA led to the following main observations. First, there was an immediate bilateral increase in the proliferation of hippocampal progenitor cells, as shown by the increased numbers of BrdU-positive cells. The increase in BrdU-positive cells was maintained over four weeks and led to increased neurogenesis, where the majority of these cells were notably double-labeled with NeuN and were present in the granular layer. Second, this increase was paralleled by improved performance in the Y-maze test, without any overt alteration in spontaneous motor behavior in the open field test. Third, both increases in neurogenesis and improvement in spatial memory were reduced to the sham basal levels by daily treatment with steroids that was administered prior to and simultaneously with the KA micro-perfusion.

Kainic acid is a well-known neuro-excitotoxic agent that has long been used to induce and study seizures in animal models of epilepsy. Previous studies, using excitotoxic doses of KA and through local injections into the brain (10 nM–20 mM) [26,27,28,29] or systemic injections (2.5–30 mg/kg) [30,31,32,33,34], have reported seizure-like behavior. In addition, several studies have documented that such KA injections into the brain result in aberrant neurogenesis and increased levels of apoptosis [34,35,36,37,38]. Nonetheless, it has been shown that intra-cerebroventricular injection of KA, at much lower doses (5 nM), can cause sustained neuronal activation without leading to seizures [39]. Even intrahippocampal injection of low doses of KA (0.037 nM–0.75 nM) does not cause seizures, but it can induce neuronal activation and might interfere with neurogenesis [12,40]. Therefore, we opted for a low dose of KA (0.5 nM/µL per hour) as an excitatory agent that was administered locally into a distant brain area with documented projection to the hippocampus [41].

Continuous micro-perfusion of low doses of KA for seven days into the thalamic AN induced an increase in DG cell proliferation at week 1, as depicted by the total number of BrdU-positive cells. This increase in BrdU-positive cells was maintained after four weeks, revealing the survival of newly born cells, and had extended to the rostral and intermediate DG of the hippocampus. Furthermore, the increase in cell proliferation and neurogenesis was noted in both the contralateral and ipsilateral sides of DG at the site of micro-perfusion. These newly born cells were exclusively located in the granular cell layer, suggesting no ectopic migration post stimulation. This was further supported by the maintained percentage of NeuN/BrdU double-positive cells in the KA groups. Therefore, this could indicate that KA administered at such low dosage into the AN might simulate sustained physiological synaptic activation of the hippocampal neurons. This comes in line with previous work showing that low sub-epileptogenic doses of KA applied directly to the DG (1 µL of 0.75 nM) elicit functional and structural re-organization [40].

Previous research on short- and long-term DBS stimulations in the same thalamic area shows almost comparable effects on hippocampal neurogenesis and spatial performance [9]. Thus, the dosage of KA micro-perfusion used in this study was enough to produce a similar effect in boosting hippocampal neurogenesis when compared to DBS, without eliciting abnormal motor behavior. Furthermore, the sustained effect of the KA micro-perfusion was comparable to that of long-term electrical stimulation in inducing a bilateral increase in cell proliferation and neurogenesis. Although using much lower doses of KA, our results appear to be in line with previous reports that show a bilateral increase in NSC proliferation following the injection of high doses of KA in a rat model of temporal lobe epilepsy [40,42,43]. In general, hippocampal neurogenesis is well regulated by afferent input via the perforant path and through the release of excitatory neurotransmitters [44]. Moreover, up to 50% of the SGZ stem/progenitor cells receive synaptic inputs [45]. Therefore, our observation of a bilateral increase in hippocampal neurogenesis might be mediated by an increased synaptic efficacy of the thalamic AN–dentate connections, which was induced by the KA micro-perfusion [46,47].

The open field test, at an early and a later time point (one and four weeks), aimed to detect whether there were any differences in the spontaneous motor activity of the KA micro-perfused rats in comparison to the sham rats (Appendix A). KA did not induce significant alteration in the observed indicators of spontaneous motor behavior when compared to the sham rats. In addition, the rats that received KA, as well as those that received KA+ dexamethasone, had similar recordings for total distance traveled, average speed, and total time spent immobile as those of the sham group. Moreover, all groups had similar exploratory behavior as they all recorded similar number of entries into the central zone of the open field and spent comparable time there at both week 1 and week 4.

The increase in DG cell proliferation and neurogenesis was associated with increased exploration of the novel arm in the Y- maze test, which was observed at weeks 1 and 4. Spatial navigation is mainly processed through neuronal circuits that link the AN with the hippocampus [41,48,49]. The AN has significant contributions to spatial processing and learning [50], possibly through the reported large numbers of “head direction cells” that play a role in spatial navigation. These specialized cells emit discharges when the animal steers its head in a certain direction [51] and maintain firing tendencies in novel environments [52,53]. Furthermore, the AN is a key relay station in the Papez circuitry that connects extensively with the hippocampal formation [54,55], and damages to either one result in a disruption of spatial performance [56,57]. In view of this, the sustained increase in neurogenesis following the KA micro-perfusion in the AN could be correlated with the higher tendency to spatially explore the novel arm compared to the other two arms.

It is well established that sustained increase in corticosteroid plasma levels simulates stress conditions and inhibits the proliferation of stem/progenitor cells in the DG of the hippocampus [19]. Stress-induced sustained increase in corticosterone level is known as a powerful suppressor of hippocampal neurogenesis. This stress-induced increase in glucocorticoid acts on target tissues through glucocorticoid receptors (GR) which are highly expressed in the hippocampus [58]. One agonist to GR is dexamethasone, which has been shown to elicit cell loss in the pyramidal layers and the dentate gyrus (DG) granules [59]. This study is the first to report that infusion of the AN in the thalamus with low doses of KA maintained hippocampal neurogenesis to near baseline sham levels in rats that were subjected to daily treatment with dexamethasone prior to and during the KA micro-perfusion. Furthermore, daily treatment of the KA rats with dexamethasone brought exploratory behavior in the Y-maze to near baseline sham levels at week 4. Thus, sustained neuronal activation by KA micro-perfusion might compensate for the suppressive effect of dexamethasone. This observation is consistent with previous reports of corticosteroid-induced reduction of neurogenesis that is reversed by DBS stimulation of the AN of the thalamus [60,61].

## 5. Conclusion

In conclusion, the results of the present study demonstrate the possible use of a method for selective activation of the anterior thalamic neurons aiming to boost hippocampal neurogenesis. The continuous micro-perfusion of thalamic neurons with low doses of KA, over several days, constitutes an alternative way to deep brain stimulation that is based on the synaptic activation of hippocampal neurons through their physiological input from the anterior thalamic neurons. The induction of a bilateral increase in neurogenesis in response to unilateral thalamic AN perfusion can be considered as a sign of plasticity induced by sustained activation of hippocampal neurons (Martin et al., 2019). Moreover, the improved performance of the rats in the Y-maze test and the rescue/preservation of the basal level of neurogenesis in the rats subjected to daily treatment of corticosteroids provide further evidence of hippocampal plasticity, which is induced by the sustained activation of anterior thalamic neurons through the continuous release of low doses of kainic acid.

## Figures and Tables

**Figure 1 cells-11-03413-f001:**
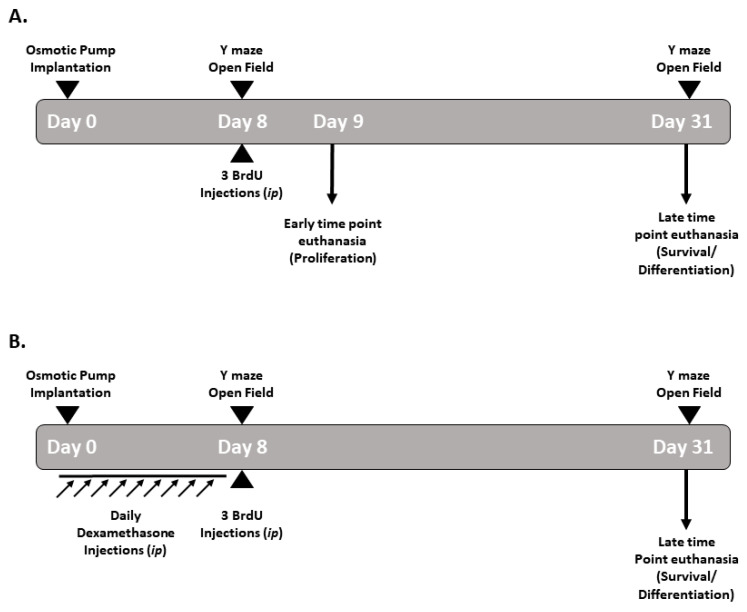
Diagrams showing the experimental timeline for continuous micro-perfusion of kainic acid using mini-osmotic pumps and other experimental procedures. (**A**) Experimental timeline for osmotic pump implantation, behavioral tests, BrdU injections and euthanasia for sham and KA groups and (**B**) for dexamethasone sham- and KA- treated groups.

**Figure 2 cells-11-03413-f002:**
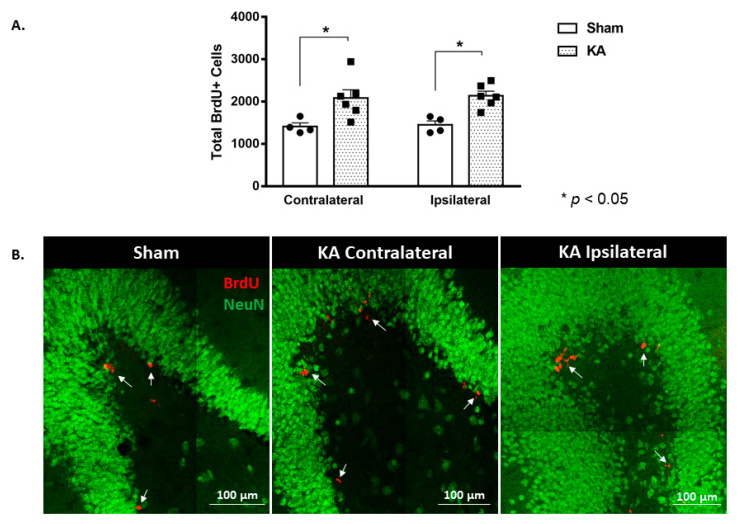
Bilateral increase in the proliferation of progenitor cells in the dentate gyrus at week 1 following unilateral continuous micro-perfusion of kainic acid into the AN. (**A**) Stereological quantification of BrdU-labeled cells in the DG of the KA micro-perfused (*n* = 6) and sham (*n* = 4) groups. Each bar represents the average ± SEM of the BrdU quantification. Two-way ANOVA followed by Tukey’s multiple comparison test was used to determine the significance between the KA micro-perfused and sham groups at both the ipsilateral and contralateral sides. (**B**) Representative confocal images showing immunofluorescence labeling of NeuN and BrdU (arrows) in the DG of the different groups. Images were taken as Z stacks using a 40×-oil objective.

**Figure 3 cells-11-03413-f003:**
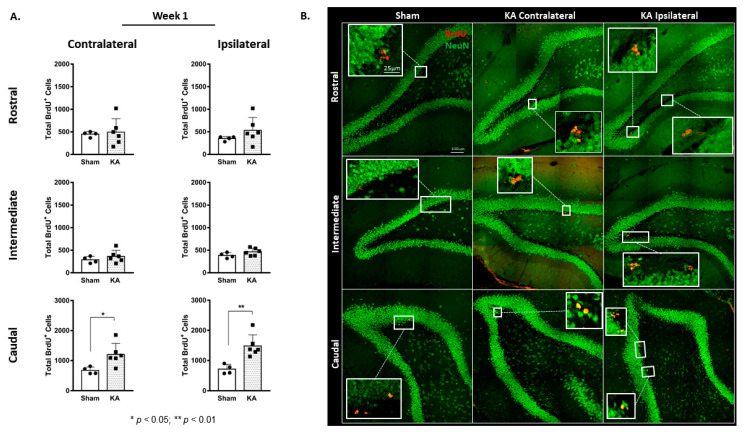
Spatial distribution of increased cell proliferation in the ipsilateral and contralateral dentate gyrus at week 1 following continuous micro-perfusion of kainic acid into the AN. (**A**) The graphs illustrate the spatial distribution of the total number of BrdU-labeled cells in the contralateral and ipsilateral DG in the sham (*n* = 4) and KA micro-perfused (*n* = 6) rats. Determination of significance was made using two-tailed unpaired Student’s *t*-test. (**B**) Confocal images illustrating BrdU-labeled cells in the rostral, intermediate, and caudal segments of the DG in the sham and KA micro-perfused rats (ipsilateral and contralateral sides with reference to the injection site). Scale bar is 100 µm.

**Figure 4 cells-11-03413-f004:**
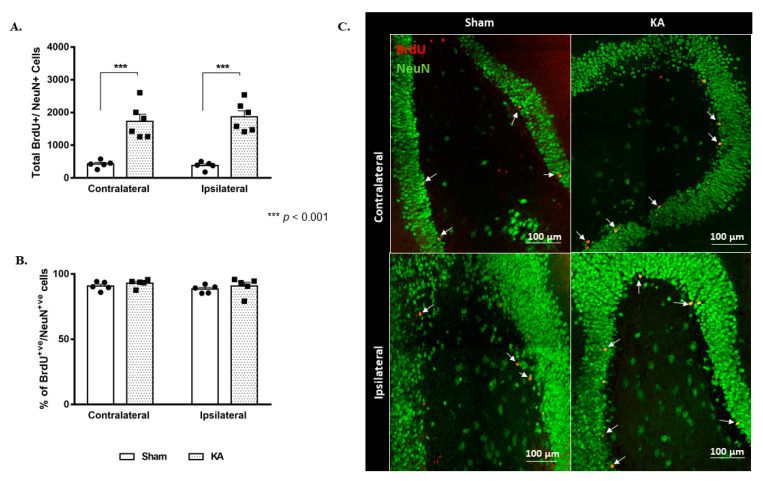
Sustained increase in neurogenesis in the ipsilateral and contralateral dentate gyrus following continuous micro-perfusion with kainic acid into the AN. (**A**) Stereological quantification of BrdU-labeled cells in the DG of the sham (*n* = 5) and KA micro-perfused (*n* = 6) groups at week 4. Determination of statistical significance of differences was made using two-way ANOVA followed by Sidak’s multiple comparison test. (**B**) Graph showing the percentage of double-positive BrdU/NeuN out of the total BrdU-positive cells in the contralateral and ipsilateral DG of the sham and KA micro-perfused rats. (**C**) Representative confocal images showing immunofluorescence labeling of NeuN and BrdU in the DG of the sham and KA micro-perfused groups at week 4. BrdU-positive cells (arrows) in the KA micro-perfused group in the ipsilateral and contralateral DG at the site of injection. Images were taken as Z stacks using a 40X-oil objective. +ve: positive.

**Figure 5 cells-11-03413-f005:**
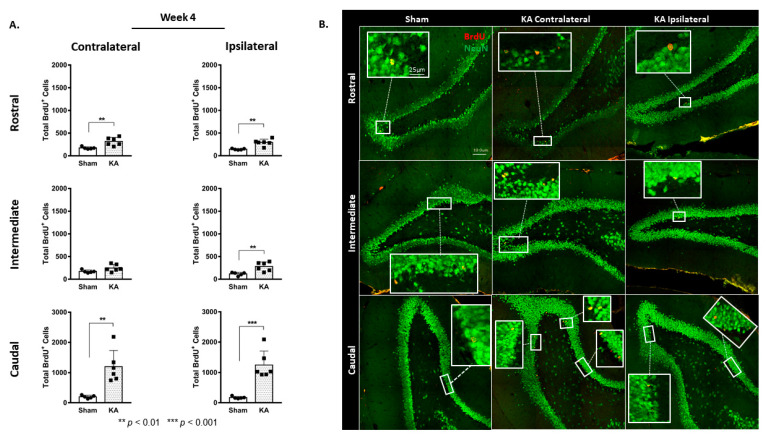
Spatial distribution of neurogenesis showing sustained increase in the ipsilateral and contralateral dentate gyrus at week 4 following continuous micro-perfusion of kainic acid into the AN. (**A**) Total number of BrdU-labeled cells in the rostral, intermediate, and caudal segments of the contralateral and ipsilateral DG in the sham (*n* = 5) and KA micro-perfused (*n* = 6) rats at week 4. Determination of significance of differences was made using two-tailed unpaired Student’s *t*-test. (**B**) Confocal images illustrating the distribution of the BrdU-labeled cells in different segments of the DG in the KA micro-perfused (ipsilateral and contralateral sides) and sham rats at week 4. Scale bar is 100 µm.

**Figure 6 cells-11-03413-f006:**
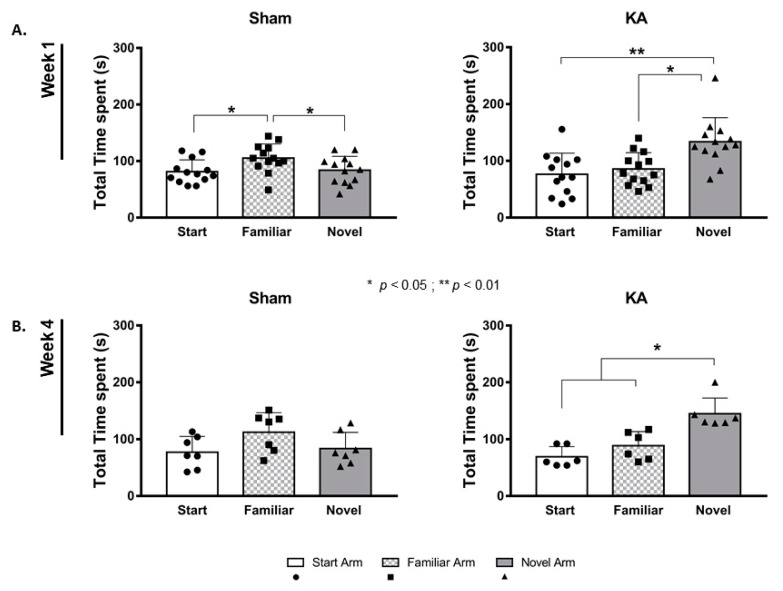
Enhanced exploratory behavior at weeks 1 and 4 following continuous micro-perfusion of Kainic Acid into the AN. Graph representing the mean ± SEM of the total time spent in the novel, familiar (second arm), and start arms. The test was performed at week 1 (**A**) and at week 4 (**B**) following the saline (*n* = 7 and *n* = 13 for week 1 and week 4, respectively) or KA micro-perfusion (*n* = 6 and *n* = 13 for week 1 and week 4, respectively). One-way repeated measures ANOVA followed by Tukey’s multiple comparison test was performed to determine statistically significant differences in the total time spent between the different arms within the same group.

**Figure 7 cells-11-03413-f007:**
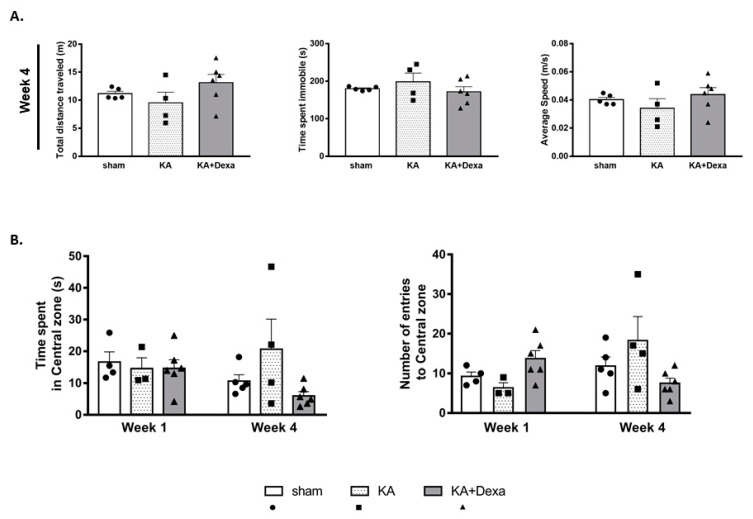
Central zone exploration and mobility in the open field were not affected by KA micro-infusion. (**A**) Graphs representing mobility measures: the total distance traveled by the sham (*n* = 5), KA (*n* = 4), and KA + Dexa (*n* = 6) rats, the total time spent immobile, and the average speed, and (**B**) exploration measures: number of entries to the central zone and total time spent in the central zone at week 1 and week 4 for the sham (*n* = 4 and *n* = 5, respectively), KA (*n* = 3 and *n* = 4, respectively) and KA + Dexa (*n* = 6 at both week 1 and week 4) rats.

## Data Availability

Not applicable.

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
