# Peer review of "Sustained Activation of the Anterior Thalamic Neurons with Low Doses of Kainic Acid Boosts Hippocampal Neurogenesis"

_cells, 2022, doi:10.3390/cells11213413_

Round 1

Reviewer 1 Report

The authors have shown the low dose of Kainic acid can promote the hippocampal neurogenesis. The manuscript needs to go through major revision before it is accepted.

1. Authors should consider presenting NeuN and BrdU channels separately for easy understanding of their co-localization.

2. In behavior studies the number of animals used in each group were not uniform, looks like these study results were conspicuously prepared. The number of animals used in figure 6 and figure 7 in each group has significant difference. It is recommended make it uniform.

3. Please mention the number of animals analyzed for the behavior study in the respective figure legend.

Author Response

Reviewer 1:

The authors have shown the low dose of Kainic acid can promote the hippocampal neurogenesis. The manuscript needs to go through major revision before it is accepted.

  1. Authors should consider presenting NeuN and BrdU channels separately for easy understanding of their co-localization.

We thank the reviewer for their constructive comments. We present here a new supplementary figure showing examples of separate channels as the reviewer suggested and we will leave it up to you whether to consider it as supplementary image 2 or not (Image in attached word document).

     Regarding presenting NeuN and BrdU channels separately, the best way to show co-localization is viewing channels together. We should clarify that at the early euthanasia timepoint the NeuN positive cells are not co-labeled with BrdU. Rather at this proliferative stage the NeuN positive cells making the granule layer are used as a reference for the neurogenic niche which is directly beneath this layer (the subgranular zone). For the groups euthanized at week 4, it is important to show the new location of the BrdU positive cells in the granular layer at this stage. Cells outside the granular layer and still in the subgranular layer are only positive for BrdU, not NeuN. Thus, the importance of co-labeling is not just to show co-localization of BrdU and NeuN positive cells but also to reflect the location of the newly born cells which changes as these cells mature over time.  Based on this, we included some examples showing zoomed in confocal images to accurately reflect location and we leave it up to the reviewers to decide whether to include this in the manuscript or not.

  1. In behavior studies the number of animals used in each group were not uniform, looks like these study results were conspicuously prepared. The number of animals used in figure 6 and figure 7 in each group has significant difference. It is recommended make it uniform.

In figure 6, please note that the number of rats in the y-maze test at week 1 is higher than that at week 4 as it includes the groups that were euthanized at week 1 and the groups that were euthanized later at week 4. As for Y-maze at week 4, it only includes the rats that would be euthanized on that week since the week 1 group had already been euthanized by this timepoint. Thus, on week 1, the numbers would be added from both groups and hence the difference in numbers between week 1 and week 4 in this graph. Please note that this is explained in the materials and methods section in Novel arm exploration part (lines 113 and 114)

As for figure 7, please note that now the number of rats at week 4 are consistent across all parameters and same for week 1. Figure 7 has been edited accordingly and replaced by a new updated figure.

  1. Please mention the number of animals analyzed for the behavior study in the respective figure legend.

Based on reviewer’s comments, please note that we have now added the number of rats included in each test in its respective figure legend (only figures 6 and 7 did not include the number of animals analyzed and thus their figure legends were edited accordingly). 

Reviewer 2 Report

Chamaa, Darwish and colleagues describe observations performed in the DG of young adult rats injected intrathalamically with low doses of Kainic acid (KA) over a period of 7 days.
The authors report increased BrdU+ cell abundance in the SGZ shortly after the termination of KA treatment, and an increased abundance of BrdU+ NeuN+ cells in the granule cell layer at 4weeks.
The effects are only observed (or most pronounced, at 4wk) within the caudal/ventral dentate domain and occur bilaterally.
Dexamethasone treatment starting 2d before (and lasting throughout) KA delivery eliminates the effects of KA, in line with the known impact of glucocorticoid signalling on adult neurogenesis.
KA-treated rats show increased exploration of the novel arm in a Y-maze test, while Dexamethasone treatment removes the effect (or rather reverses it, in relation to the familiar arm, when compared to the KA-only treatment). Open field tests show increased inter-individual variability in terms of mobility for both KA- and KA/Dexamethasone-treated animals, without other clear trends.

The experimental approach is clearly described and the results show low-dose KA-based stimulation of an afferent region can be used to boost region-specific adult hippocampal neurogenesis, at least in the short term, as an alternative to other interventions.

Here are a few points I think should be addressed

1-      If material is available, the authors could assess the abundance of proliferating (e.g. Ki67+) cells within the SGZ at 1 and 4wks. Even alone (i.e. not in combination with NSC-specific markers), this may indicate if, 3 weeks after the termination of KA administration, any effect on the NSC lineage is persisting. More complex analyses are probably beyond the scope of this work.

-      In lines 271-271 (and similarly in 286-287, both in the Discussion), the authors claim “an immediate bilateral increase in the proliferation of hippocampal progenitor cells was maintained over four weeks”, which at present is unsupported, since the 4wk data only refer to NeuN+ cells and no info is available as to the ongoing behaviour of neural progenitors.

3-      In lines 272-275, the authors suggest performance in the Y-maze shows “improvements in spatial memory”. I am no behavioural expert, but I wonder if reversal of preferential exploration from the familiar (sham animals) to the novel branch (KA animals) can be taken to mean much about memory, vs e.g. more specifically novelty-seeking. The effect of Dexamethasone, with the increased exploration of the familiar branch and strongly reduced exploration of the novel branch, suggests that the novel branch is recognized/remembered as novel/previously inaccessible but avoided, which to me suggests a somehow intact memory.

4-      In the discussion (lines 297-298) the authors state that the dosage of KA was “enough to produce neuronal activation comparable to physiological conditions”, without having shown how much neuronal activation (where?) is actually induced by KA. The claim seems to rest on the _assumption_ that any impact on DG neurogenesis only reflects neuronal activation, so it seems somehow circular.

Author Response

Reviewer 2:

  • If material is available, the authors could assess the abundance of proliferating (e.g. Ki67+) cells within the SGZ at 1 and 4wks. Even alone (i.e. not in combination with NSC-specific markers), this may indicate if, 3 weeks after the termination of KA administration, any effect on the NSC lineage is persisting. More complex analyses are probably beyond the scope of this work.

We thank the reviewer for their valuable comments. We appreciate the suggestion of ki67 staining and think it would add further value and answer important questions in this narrative. However, with the constraints of time and material, we are unable to perform this quantitative analysis in such a restricted time window.

  • In lines 271-271 (and similarly in 286-287, both in the Discussion), the authors claim “an immediate bilateral increase in the proliferation of hippocampal progenitor cells was maintained over four weeks”, which at present is unsupported, since the 4wk data only refer to NeuN+ cells and no info is available as to the ongoing behaviour of neural progenitors.

We thank the reviewer for pointing out this important remark. We have now edited these statements in li es 271 and 286 to make a clear differentiation between proliferation at early timepoint and tracking BrdU positive cells at week 4.

  • In lines 272-275, the authors suggest performance in the Y-maze shows “improvements in spatial memory”. I am no behavioural expert, but I wonder if reversal of preferential exploration from the familiar (sham animals) to the novel branch (KA animals) can be taken to mean much about memory, vs e.g. more specifically novelty-seeking. The effect of Dexamethasone, with the increased exploration of the familiar branch and strongly reduced exploration of the novel branch, suggests that the novel branch is recognized/remembered as novel/previously inaccessible but avoided, which to me suggests a somehow intact memory.

We thank the reviewer for this important comment. The type of memory tested in the Y-maze is referred to as “spatial reference memory”, which refers to the rodents’ ability to spatially recognize a novel previously unexplored environment (Kraeuter et al., 2019; Prieur & Jadavji, 2019). Here, the data indicate that the KA rats were more capable of recognizing the previously unexplored arm and thus were able to denote its novelty by spending more time exploring it as compared to familiar and start arm.

As for Dexa-treated KA rats, their pattern of exploration was similar to that of sham rats, however, this doesn’t refer to intact memory as these rats are treated with KA and the effect of KA in boosting exploration was no longer evident which could mean that there has been an effect on the rats’ spatial memory that reverted their exploratory behavior back to sham baseline levels. Nonetheless, in order to say intact, we would be assuming that KA had no effect in boosting exploration and that dexamethasone had no impact in decreasing exploration.

4-      In the discussion (lines 297-298) the authors state that the dosage of KA was “enough to produce neuronal activation comparable to physiological conditions”, without having shown how much neuronal activation (where?) is actually induced by KA. The claim seems to rest on the _assumption_ that any impact on DG neurogenesis only reflects neuronal activation, so it seems somehow circular.

We totally agree with the reviewer on this remark. We did not test for neuronal activation aside from showing increased neurogenesis, thus we have edited the mentioned statement in lines 297-298 to be more accurate.

Kraeuter, A. K., Guest, P. C., & Sarnyai, Z. (2019). The Y-Maze for Assessment of Spatial Working and Reference Memory in Mice. Methods in molecular biology (Clifton, NJ), 1916, 105-111. doi:10.1007/978-1-4939-8994-2_10

Prieur, E. A. K., & Jadavji, N. M. (2019). Assessing Spatial Working Memory Using the Spontaneous Alternation Y-maze Test in Aged Male Mice. Bio Protoc, 9(3), e3162. doi:10.21769/BioProtoc.3162

Round 2

Reviewer 1 Report

I am satisfied with author response. I do not have any further comments. The manuscript can be accepted for publication

Reviewer 2 Report

The authors have satisfactorily addressed my few concerns regarding neurogenesis-related effects.

I appreciate the authors´ explanation (and refs in support) of their interpretation of the y-maze results. I still wonder if the test actually allows distinguishing between memory performance and novelty-seeking (which requires anyway memory of previous exposure); i.e. I don´t see how one can tell a rodent with better memory from one with intact memory but higher drive towards exploring a novel spot. Maybe looking at time to initiation of exploration and time spent exploring novel arms may provide clues in this sense. In any case, as I have come across contrasting reports regarding the impact of adult hippocampal neurogenesis on novelty-seeking (e.g. refs in 10.3389/fnins.2022.852680), I am not expecting the authors to discuss this further.